# Characterization of *Leptoglossus occidentalis* Eggs and Egg Glue

**DOI:** 10.3390/insects14040396

**Published:** 2023-04-19

**Authors:** Eva Sánchez-Hernández, Pablo Martín-Ramos, Jonatan Niño-Sánchez, Sergio Diez-Hermano, Flor Álvarez-Taboada, Rodrigo Pérez-García, Alberto Santiago-Aliste, Jesús Martín-Gil, Julio Javier Diez-Casero

**Affiliations:** 1Department of Agricultural and Forestry Engineering, ETSIIAA, University of Valladolid, Avenida de Madrid 44, 34004 Palencia, Spain; alberto.santiago@estudiantes.uva.es (A.S.-A.); jesus.martin.gil@uva.es (J.M.-G.); 2Instituto Universitario de Investigación en Gestión Forestal Sostenible (iuFOR), Universidad de Valladolid, Avenida de Madrid 57, 34071 Palencia, Spain; jonatan.nino@uva.es (J.N.-S.); sergio.diez.hermano@uva.es (S.D.-H.); rodri.pg1993@gmail.com (R.P.-G.); juliojavier.diez@uva.es (J.J.D.-C.); 3Departamento de Producción Vegetal y Recursos Forestales, ETSIIAA, Universidad de Valladolid, Avenida de Madrid 57, 34071 Palencia, Spain; 4School of Agrarian and Forest Engineering, DRACONES, Universidad de León, Avenida de Portugal 41, 24401 Ponferrada, Spain; flor.alvarez@unileon.es

**Keywords:** eggs, GC–MS, FTIR, glue, semiochemicals, SEM-EDS, oviposition, western conifer seed bug, WCSB, pine nuts, edible pine

## Abstract

**Simple Summary:**

This study explored the chemical components of the egg glue used by the Western Conifer Seed Bug (*Leptoglossus occidentalis* Heidemann, 1910) to agglutinate eggs and adhere to pine needles. Results showed that the adhesive secretion includes plasticizers and thermoplastic elastomer resins with semiochemical properties in an oily matrix containing proteins. This knowledge of the egg glue composition can be used to develop new control strategies for *L. occidentalis*, potentially limiting the economic impact caused by this pest insect that reduces the production of pine nuts by up to 25%.

**Abstract:**

The western conifer seed bug (*Leptoglossus occidentalis* Heidemann, 1910, Heteroptera: Coreidae) has a significant economic impact due to the reduction in the quality and viability of conifer seed crops; it can feed on up to 40 different species of conifers, showing a clear predilection for *Pinus pinea* L. in Europe. Its incidence is especially relevant for the pine nut-producing industry, given that the action of this pest insect can reduce the production of pine nuts by up to 25%. As part of ongoing efforts aimed at the design of control strategies for this insect, this work focuses on the characterization (by scanning electron microscopy–energy-dispersive X-ray spectroscopy, Fourier-transform infrared spectroscopy, and gas chromatography–mass spectroscopy, GC–MS) of the compounds released by these insects during oviposition, with emphasis on the adhesive secretion that holds *L. occidentalis* eggs together. Elemental analysis pointed to the presence of significant amounts of compounds with high nitrogen content. Functional groups identified by infrared spectroscopy were compatible with the presence of chitin, scleroproteins, LNSP-like and gelatin proteins, shellac wax analogs, and policosanol. Regarding the chemical species identified by GC–MS, eggs and glue hydromethanolic extracts shared constituents such as butyl citrate, dibutyl itaconate, tributyl aconitate, oleic acid, oleamide, erucamide, and palmitic acid, while eggs also showed stearic and linoleic acid-related compounds. Knowledge of this composition may allow advances in new strategies to address the problem caused by *L. occidentalis*.

## 1. Introduction

### 1.1. On Compounds Released by Insects

The secretions of insects and the effects that they trigger in host plants have been the subject of research since the early 2000s [1]. Some of these secretions fulfill a defensive function (e.g., osmeterial secretions), others convey specific chemical messages that modify behavior or physiology (e.g., semiochemicals), but most are involved in egg attachment.

Osmeterial-like secretions (which are odoriferous liquids secreted by epidermal glands present in the prothoracic segment of papilionid larvae when the larva feels threatened) contain monoterpene hydrocarbons, sesquiterpenic compounds, or a mixture of aliphatic acids and esters: isobutyric acid, 2-methylbutyric acid, and their methyl esters, methyl 3-hydroxybutanoate, 3-hydroxybutanoic acid, α-pinene, myrcene, limonene, 2-methylpropanoic acid, 2-methylbutanoic acid, and their methyl and ethyl esters.

Semiochemicals are organic compounds used by insects to locate the potential host community (kairomones) or for attractive sexual purposes (pheromones). For example, *Phthia picta* (Drury, 1773) male-produced sex pheromone contains 5,9,17-trimethylhenicosane [2], but, in some cases, the male-released *trans*-2-hexenyl *trans*-2-hexenoate, *trans*-2-hexenyl *cis*-3-hexenoate, and myristyl isobutyrate play the double role of attractant pheromone for adults as well as aggregation pheromone.

Concerning extracts of egg clusters (such as those from *Telenomus podisi* Ashmead, 1893), previous studies showed that the major components were saturated and unsaturated fatty acids, including hexadecanoic acid, linoleic acid, and *cis*-9-octadecenoic acid [3]. In regards to egg glues, hydrogel glues proved to be rich in glycine, serine, and/or proline, and some contained substantial levels of 4-hydroxyproline [4]. Egg glue proteins (EGPs), produced by female insects, can make the eggs firmly attached to the oviposition sites. For instance, the full-length sequence and secondary structure of silkworm EGP are that of a repetitive pentapeptide motif, G-G-N/K/D-Q/E/K-Q/P, repeated 346 times, forming a hydrophilic and elastic *β*-spiral structure [5].

The most complex fluid composition an insect can produce is one that comprises four key components: an aqueous portion rich in amino acids and carbohydrates; oily nanodroplets containing hydrocarbons; an emulsifier (e.g., cholesterol, monoglycerides, etc.) to stabilize the mixture; and volatile organic compounds like kairomones or pheromones.

### 1.2. Description and Importance of Leptoglossus occidentalis

The western conifer seed bug, *Leptoglossus occidentalis* Heidemann (1910) (Hemiptera: Coreidae), is a pest that causes significant damage to *Pinus pinea* L. nut production (one of the most profitable non-wood forest products in several Mediterranean countries) [6,7] and to other host conifers such as *Pinus contorta* Douglas ex Loudon, *Pinus glauca* Moench, and *Pseudotsuga menziesii* (Mirb.) Franco. Both nymphs and adults feed on seeds by piercing their stylets into cones, causing conelet abortion and seed loss. Based on previously determined damage estimates for each life stage of feeding during three periods of cone development, a hypothetical density of one seed bug per tree early in the season will result in an expected seed loss of approximately 310 seeds [8]. Due to globalization, this pest has colonized almost all European countries and beyond, including marine islands, from Portugal to the south of Russia and Kazakhstan, and from Norway to the Maghreb, Turkey, Lebanon, and the Syrian Golan Heights, as well as other areas of the world such as Japan, Korea, Chile, or Uruguay [9]. The ruin of businesses by this pest has led many companies and self-employed to abandon the activity.

Phenology is dependent on weather conditions, which may vary between years. The life cycle of the western conifer seed bug in the Mediterranean area has two annual generations, with the first laying in mid-May and the second in August [10]. According to Barta [11], a single female in the field can lay up to 30 eggs during her life cycle; in laboratory observations, clutches range from 9 to 16 eggs in a row. Eggs are laid on pine needles, from which the first-instar nymphs emerge. After emergence, the newly hatched nymphs remain together on a shoot close to where the clutch is located [12]. Young nymphs are orange and brown, turning reddish-brown as they develop [13]. Approximately 9% of them survive until adulthood [8]. *L. occidentalis* nymphs go through five instar stages, which last between one and three weeks, before transforming into the first generation of adults (mid-July). These adults lay further clutches, which give rise to the second generation of nymphs during the second half of the summer and to new adults from late August and September onward. These adults do not reproduce and overwinter in groups until the following spring [10].

Environmentally friendly management of this species is the main objective of the different research groups working with *L. occidentalis*. Recently, a new Tachinid parasitizing western conifer seed bugs was found in Portugal [14]. However, the biological control of this species is still far away. Under this scenario, the use of semiochemicals for the management of this species may be a promising strategy for the control of this pest. To the best of our knowledge, there is no effective trapping technique for *L. occidentalis*. Traps with different colors and conformations were tested, but all performed poorly to attract this bug [15], and although *L. occidentalis* males produce an aggregation pheromone to which both males and females respond [16,17,18], it has not yet been identified. However, important advances have been recently made by Millar et al. [19]: leptotriene (i.e., 1*S*,8*R*,*E*)-9,9-dimethyl-2-methylene-5-vinylbicyclo[6.2.0]dec-4-ene, a unique sesquiterpene hydrocarbon produced by sexually mature males of the leaffooted bugs *Leptoglossus zonatus* (Dallas, 1852) and *L. occidentalis,* was recently described and identified by electroantennogram response.

Applications of dimethoate or carbaryl, used to control other insects, have been suggested as possible treatments for the control of conifer seed bugs. Another control option, using traps that emit infrared light wavelengths, was explored by researchers at Simon Fraser University and the British Columbia Ministry of Forests and Range [20]. The traps that achieved the best results for *L. zonatus*, a relative species of *L. occidentalis*, were green and blue, according to Franco-Archundia et al. [21], and yellow ones with a polytetrafluoroethylene (Fluon^®^) lubricant coating [22]. In a recent article by Inoue et al. [23], the identification of the alarm and sex pheromones from nymphs and adults of the leaf-footed bug, *L. zonatus*, was elucidated as 1-hexanol, hexanoic acid, hexyl acetate, octyl acetate, 5-ethyl-2(5H)-furanone, *cis*-*β*-ocimene, *cis*-allo-ocimene, decanal, *α*-*trans*-bergamotene, and *trans*-*β*-farnesene.

### 1.3. Aim of the Study

As part of ongoing efforts aimed at the design of control strategies for the western conifer seed bug, the work presented herein attempts to: (1) characterize the egg attachment glue of *L. occidentalis*; (2) identify volatile compounds in *L. occidentalis* eggs analogous to those of *L. zonatus*; and (3) determine if any odorous volatile component of pine bark or pine nuts can be related to the unknown secretion released by *L. occidentalis*. This information may provide leads that can be developed for the control of this insect.

## 2. Material and Methods

### 2.1. Egg and Egg Glue Samples

Fresh eggs of *L. occidentalis* were used for this study. The eggs and glue samples were collected from the pine needles of 2-3-year-old potted *Pinus halepensis* Mill. saplings (provided as ovipositional substrate) at the Forest Pathology Lab at the University of Valladolid (Palencia, Spain), where the western conifer seed bugs are reproduced and maintained in cages under controlled laboratory conditions at a temperature of 21 ± 2 °C, a relative humidity of 40 ± 10%, and under daylight photoperiod. The individuals were housed in rearing boxes measuring 47.5 cm × 47.5 cm × 93 cm, with a transparent plastic front panel with two openings for manual handling and side walls made of 1 mm × 0.3 mm light polyester mesh. *Ad libitum* shelled *P. pinea* kernels were provided as food [19].

The aforementioned colony was reared from adults collected from a heavily infected *P. pinea* forest in the province of Valladolid (‘El Molinillo’, Tordesillas, Valladolid, Spain).

Needle removal from the potted pine saplings was carried out during the months of May and June, selecting needles with fully hatched clutches. For this purpose, the presence of an operculum due to the nymph’s exit (or a color change indicating the abortion of the embryo) was checked for each egg. The selected needles were cut using scissors, placed in Petri dishes, and sealed with Parafilm.

### 2.2. Characterization

The morphology and multi-elemental composition of the egg and glue samples from the abovementioned colony reared in captivity were analyzed by scanning electron microscopy with energy-dispersive X-ray spectroscopy (SEM-EDX), using an EVO HD 25 (Carl Zeiss, Oberkochen, Germany) apparatus. Measurement conditions are indicated in each micrograph.

The infrared spectra were recorded using a Nicolet iS50 (ThermoFisher Scientific; Waltham, MA, USA) FTIR spectrometer with a diamond attenuated total reflection (ATR) module. Operative conditions were: room temperature; a 400–4000 cm^−1^ range; a 0.5 cm^−1^ spectral resolution; and the co-addition of 64 scans.

Egg/egg glue samples were mixed in a 1:20 (*w*/*v*) ratio with a methanol/water solution (1:1 *v*/*v*) and heated in a water bath at 50 °C for 30 min. The mixture was then sonicated in pulse mode for 5 min with a 1-min stop every 2.5 min, using a model UIP1000hdT probe-type ultrasonicator (Hielscher Ultrasonics; Teltow, Germany). The solution was centrifuged at 9000 rpm for 15 min, and the supernatant was filtered through Whatman No. 1 paper before being freeze-dried to obtain the solid residue. For subsequent gas chromatography-mass spectrometry (GC–MS) analysis, 25 mg of the freeze-dried extract was dissolved in 5 mL of HPLC-grade methanol (gradient grade, suitable as an ACS-grade LC reagent, ≥99.9; Sigma-Aldrich) to obtain a 5 mg·mL^−1^ solution, which was further filtered.

The GC–MS characterization was outsourced to the University of Alicante’s Research Support Services (STI) and was conducted using a model 7890A gas chromatograph coupled to a model 5975C quadrupole mass spectrometer (both from Agilent Technologies; Santa Clara, CA, USA). The chromatographic conditions were as follows: 1 µL injection volume; 280 °C injector temperature; splitless mode; 60 °C initial oven temperature, kept for 2 min, followed by a 10 °C·min^−1^ ramp up to a final temperature of 300 °C, maintained for 15 min. For the separation of the compounds, an Agilent Technologies HP-5MS UI chromatographic column (30 m in length, 0.250 mm in diameter, and 0.25 µm film) was used. In regards to the mass spectrometer conditions, the temperature of the electron impact source of the mass spectrometer and the quadrupole were 230 and 150 °C, respectively, with a 70 eV ionization energy. For the identification of components, their mass spectra and retention times were compared with those of the authentic compounds and with the database of the National Institute of Standards and Technology. Siloxane-derivatives, which may arise from contamination from a variety of sources such as inlet septa, columns, valves, solvents, and vial cap seals, represented less than 0.9% of the peak area in the egg and glue sample chromatograms and were removed from the identified phytochemical lists.

## 3. Results and Discussion

### 3.1. SEM−EDS Characterization

Scanning electron micrographs of both the hemicylindrical eggs, which are laid in long chains, end to end, along the needles of the host tree [24], and the egg glue are presented in Figure 1. Figure 1a shows an intact egg, while Figure 1b shows a hatched egg. Figure 1c shows the egg glue.

The EDS analysis results are summarized in Table 1, along with the elemental composition of several potentially related organic chemicals.

### 3.2. Vibrational Characterization of the Eggs and Their Glue

The absorption bands of *L. occidentalis* eggs and glue (as shown in Table 2) correspond to functional groups compatible with the components found in the cuticle (chitin and scleroproteins), Louse Nit Sheath Protein (LNSP)-like and gelatin proteins, shellac wax analogs, and related products (such as policosanol). These components are all essential for maintaining the viability of the egg, besides the adhesive function.

The prediction of protein secondary structure from FTIR spectra typically relies on the amide I–amide II region, specifically the band at approximately 1630 cm^−1^. The characterization of the nit sheath of the human head louse (*Pediculus humanus* subsp. *capitis* de Geer, 1778) reveals that the glue consists mainly of proteins, such as LNSP 1 and 2 [25], which predominantly contain Gly (25.4%), Glx (Glu or Gln, 24.4%), Ala (20.2%), and Val (10.0%) residues. In the amide spectral range, arginine has a strong absorption near 1673 cm^−1^ that is assigned to *ν*_as_(CN_3_H_5_^+^), a less strong one near 1633 cm^−1^ that is assigned to *ν*_s_(CN_3_H_5_^+^), and a weaker one at 1522 cm^−1^ that is assigned to *δ*_s_(CN_3_H_5_^+^). Furthermore, the deconvoluted amide I bands are centered around 1610, 1626, 1651, and 1691 cm^−1^.

The band of gelatin is generally located at 1630 cm^−1^, and its shift to 1634 cm^−1^ may indicate complexation, such as through the grafting of a polypeptide chain (gelatin) onto chitin. Other bands, such as those at 1445 and 1156 cm^−1^, appear to correspond to PAla.

Bands at 2922, 2858, 1729, 1455, 1226, 1173, 1015, and 729 cm^−1^ have been previously associated with policosanol-rich insect waxes [26,27], such as shellac, and with saffron [28]. Saffron’s composition includes carotenoids, safranal, and primarily lanierone (2-hydroxy-4,4,6-trimethylcyclohexa-2,5-dien-1-one), which serves as both an odorous volatile component of saffron and a pheromone emitted by the pine engraver (*Ips* spp.).

**Table 2 insects-14-00396-t002:** Main infrared absorption bands of western conifer seed bug eggs and egg glue and possible assignments, together with bands reported in the literature (for comparison purposes).

Eggs	Egg Glue	Gum Moth Eggs Glue [4]	Sorghum Bugs Gelatin [29]	Shellac Wax [27]	Saffron [28]	Assignment
	3422 3386				3365	O-H stretching in alcohols
3270	3269		3274	3269		N-H stretching/typical of geraniol
3068	3106	3081				hydrocarbon chains in polymeric films/ =C-H stretching in aromatics
2965	2960	2953		2955		C-H methoxyl, cuticles (chitins)
2922	2918		2927	2916	2924	CH_2_– asymmetrical stretching, in butyl itaconate/ COCH_3_, in chitins/policosanol
2858	2849			2850	2854	CH_2_– symmetrical stretching, in policosanol and cuticles (chitins)
1729	1736			1738	1741	C=O stretching, in policosanol, fatty acids, and lipids/C=C, phorone
1634	1626	1653	1633		1653	amide I (C=O of N-acetyl group), in gelatin and chitins
	1547	1544	1542		1543	C–N stretching/amide II (N-H bend), in chitins and chitosan
1515	1472					aromatic squeletal vibrations
1455	1463 1445			1462	1462	C–H bending (scissor)/aromatic –C=C stretching, in PAla and policosanol
1385	1377	1378	1399	1374	1400 1376	–OH bending/-C-O-H in-plane bending/–CH_3_ out-of-plane bending/CH_2_– wagging and twisting/in chitins
	1308				1317	CH_3_ wagging, in amide III, proteins
1226	1245	1265		1246	1227	C-O-C chitosan/C-O polyols
	1173			1173		C(O)O policosanol
	1156	1150		1148	1157	C-O stretch in C-O-C linkages)/-C-H in chitosan and PAla
	1115			1115	1119	asymmetric in-phase ring stretching
	1073	1065	1067	1081	1075	C-H bending/C–O–C asym. stretching, in saccharide rings
1015	1028		1042	1029	1020	C-OH stretching/aromatics, phenols
	953	945		943	945	CH_3_ wagging along the chain
		837	862			*trans* =C–H out-of-plane bending
	759 729	759		720	740	*cis* =C-H out-of-plane bending/C=C in aromatic structures

### 3.3. GC–MS Characterization of Chemical Species Present in Eggs and Their Glue

In the hydroalcoholic extracts of eggs (Appendix A), the organic compounds listed in Appendix A were identified by GC–MS. The major components were tributyl acetylcitrate (33.3%) and butyl citrate (7.3%); 9,12-octadecadienoic acid (linoleic acid) and its methyl ester (25.6%); n-hexadecanoic acid (palmitic acid) and its methyl ester (6%); 1-propene-1,2,3-tricarboxylic acid tributyl ester (tributyl aconitate) (6%); octadecanoic acid (stearic acid) (4.9%); dibutyl itaconate (3.8%); and 9-octadecenoic acid (oleic acid) methyl ester (2.7%). Additionally, secondary constituents such as 9-octadecenamide (oleamide), *cis*-13-docosenamide (erucamide), squalene, and heptaethylene glycol were identified. Sex-specific elicitors such as nonanal, decanal, ocimene, caryophyllene, farnesene, and leptotriene [19] were not detected.

The egg glue hydroalcoholic extract was analyzed by GC-MS (Appendix A), and the organic compounds identified are listed in Appendix A. Butyl citrate (61.4%) was found to be the major component, followed by heptaethylene glycol (6.5%), dibutyl itaconate (4.7%), 1-propene-tributyl aconitate (3.8%), and oleic acid (3.2%). The minor components identified were oleamide, erucamide, dianhydromanitol, squalene, adipic acid, 2-ethylhexyl isobutyl ester, and palmitic acid.

Based on the information provided, it appears that eggs and egg glue share many chemical species in common, as shown in Figure 2. These include butyl citrate, dibutyl itaconate, tributyl itaconate, oleic acid (or its methyl ester), oleamide, erucamide, squalene, palmitic acid, and heptaethylene glycol. However, eggs also contain stearic and linoleic acid-related compounds that are not present in egg glue.

## 4. Discussion

### 4.1. On Elemental Analysis Results

In a first approximation, if a simplified binary mixture approach is chosen and some constituents identified by other techniques are taken as a reference, the C and O data for eggs (Table 1) could be justified by, for instance, a combination of oleic acid (ca. 84%) and chitin or 4-hydroxyproline (ca. 16%). Additionally, the data from egg glue would be compatible with a tentative combination of ca. 60% gelatin and ca. 40% oleic acid or ca. 60% arginine and ca. 40% oleic acid. However, it should be noted that the actual combinations would not be binary but would involve multiple nitrogenated and non-nitrogenated constituents. In any case, apart from speculative considerations, elemental analyses demonstrated that the egg shells and egg glue contained a significant proportion of nitrogen, indicating the presence of peptides, proteins, or other compounds with high nitrogen content.

### 4.2. On Compounds Identified by GC−MS

Concerning the accuracy of the identification of the compounds, it is worth noting that methyl esters, such as hexadecanoic acid methyl ester, 9,12-octadecadienoic acid methyl ester, and 9-octadecenoic acid methyl ester, could be artifacts associated with the use of methanol as a solvent for GC−MS analyses [30].

Regarding the known functions of the chemical species identified by GC−MS (Appendix A), several of them are plasticizers. Tributyl citrate and tributyl acetylcitrate are hydrophobic plasticizers and lacquer-forming agents [31]. 1-Propene-1,2,3-tricarboxylic acid tributyl ester (tributyl aconitate) is a hydrolysis product of tributyl acetylcitrate [32], while dibutyl itaconate is used for making plasticizers, adhesives, lubricants, and gelation accelerators [33]. They all exhibit high plasticization efficiency, excellent intermiscibility, outstanding light, water, and cold resistance, low volatility, great ductility, strong weatherability, and good biodegradability. On the other hand, other organic compounds identified in *L. occidentalis* eggs and egg glue have been previously identified as semiochemicals. High levels of *cis*-13-docosenamide (an elastomer resin) were found on the surface of females of the lone star tick, *Amblyomma americanum* L. [34]. Hexadecanoic acid methyl ester is known to be attractive to female moths of *Ostrinia furnacalis* Guenée, 1854 [35], while oleic acid, the second most abundant fatty acid in pine nut oil, is a necromone across arthropod taxa [36]. Additionally, *cis*-9-octadecenamide has been reported to be an oviposition stimulant and attractive to female moths of *O. furnacalis* [35]. Concerning palmitic, oleic, and linoleic acids, they have been previously reported in extracts of egg clusters of *T. podisi* [3].

With regards to the detection of the aforementioned ‘plasticizers’, initially, reasonable doubts arose as they were generally considered non-natural products resulting from contamination. However, they were not present in black controls (ruling out contamination from the extraction and injection methods used), and in additional GC-MS analyses conducted for new samples, their presence was confirmed. Unlike other plasticizers, such as phthalate esters that may result from solvent storage in plastic containers, their presence as contaminants in extracts has not been referred to in the review paper by Venditti [30]. It should be noted that butyl-related compounds, such as isobutyric and 2-methylbutyric acids, and their methyl and ethyl esters, have been previously reported in osmeterial secretions from the swallowtail butterfly (*Papilio memnon heronus* Frunstorfer) [37] and of papilionid larvae in the genera *Luehdorfia, Graphium,* and *Atrophaneura* [38], thus a natural origin may be advocated. Aldrich et al. [39] reported that gypsy moth caterpillars (*Lymantria dispar* L.) secretions include biogenic amides (e.g., γ-aminobutyric acid), short-chain hydroxy acids (α-hydroxyisobutyric acid), and Krebs cycle acids (e.g., isocitric acid). In the compounds identified herein, apart from citric acid, other two Krebs cycle intermediates, namely aconitate and itaconate, would be involved [40], thus supporting the natural origin of butyl citrate, dibutyl itaconate, and tributyl aconitate. Regarding the possible role these compounds may play, it is reasonable to assume that they would promote adhesion to pine components such as pine needles, cones, nutshells, etc.

### 4.3. Comparison with Pine Constituents

In their study, Liu and Xu [41] identified forty-eight volatile chemicals in pine nut shells using GC–MS, of which the following are particularly relevant: α-pinene (12.4%); β-pinene (1.8%); oleic (33.1%), palmitic (13.6%), and stearic (7.2%) acids; 3-carene (1.5%); and 1-methy-4-(1-methylethenyl)-(S)-cyclohexene (7.9%). Pinene was not found in *L. occidentalis* samples, but oleic and palmitic acids were detected in *L. occidentalis* eggs, egg glue, and pine nuts, and stearic acid was present in eggs and pine nuts. The presence of linoleic acid in the eggs is consistent with the findings of Kadri et al. [42], who reported a 30.7% content in *P. pinea* seeds (used as a food source in our experiments), as well as the occurrence of cis-oleic, linolenic, palmitic, and stearic acids. Should the composition of the ovipositional substrate (*P. halepensis* saplings) be considered instead, a recent review paper by El Omari et al. [43] on the phytochemistry of this species showed that the fatty acid profile varies depending on the analyzed plant part (e.g., seeds, cones, branches, and needles [42,44,45,46]), geographical provenance, and extraction procedure. However, there is general agreement that the main unsaturated fatty acids are linoleic and oleic acids, and that the main saturated fatty acid is palmitic acid, whereas stearic, succinic, azelaic, linolenic, decanoic, myristic, and arachidic acids would be minor constituents.

### 4.4. Usefulness of the Obtained Results

The combination of reduced penetration through the eggshell (which acts as an excellent barrier to insecticides, fungal pathogens, and some fumigants) and pesticide resistance (given that eggs treated with insecticides repeatedly have been shown to produce high levels of enzymatic activity that break down insecticides) makes eggs an extremely difficult life stage to kill for both agricultural and urban pest species [47].

While the usual management strategy is to wait for the eggs to hatch and then treat the nymphs that emerge because they are easier to kill, the reported data may find application in the development of alternative management strategies, with associated advantages (e.g., avoiding reinfestation due to eggs left behind that were not killed by the treatment, the possibility of using repellants or solvents as non-toxic means of control, etc.).

Hence, based on the reported information regarding the composition of the glue (which consists of oily and proteinaceous components with plasticizers as minority constituents), other possible control strategies (besides spraying with a neonicotinoid/pyrethroid mix) may be envisaged to attack the *L. occidentalis* egg glue system. These include spraying needles with proteases dispersed in a fatty acid medium and/or hydrolyzing the plasticizers using dehydrogenases (given that, according to Bruns and Werners [48], tributyl citrate and tributyl acetyl citrate can be hydrolyzed to citric acid and butanol).

Another relevant topic for follow-up studies would be subjecting the identified material to bioassays to determine its semiochemical properties.

## 5. Conclusions

The vibrational and GC–MS characterization results suggest that the glue system used by *L. occidentalis* females to agglutinate eggs and adhere to pine components includes plasticizers (such as butyl citrate, dibutyl itaconate, and tributyl aconitate) and thermoplastic elastomer resins with semiochemical properties (such as *cis*-13-docosenamide and *cis*-9-octadecenamide) in an oily matrix (with oleic, palmitic, and stearic acid) and proteinaceous components. Knowledge of this composition paves the way to the development of new *L. occidentalis* control strategies during the egg stage by using proteases dispersed in fatty acid media and/or dehydrogenases to attack the egg glue.

## Figures and Tables

**Figure 1 insects-14-00396-f001:**
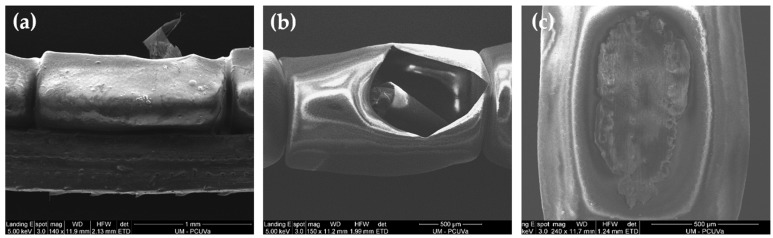
Scanning electron micrographs of *L. occidentalis* eggs and egg glue: (**a**) an egg on a pine needle; (**b**) a hatched egg; (**c**) egg glue.

**Figure 2 insects-14-00396-f002:**
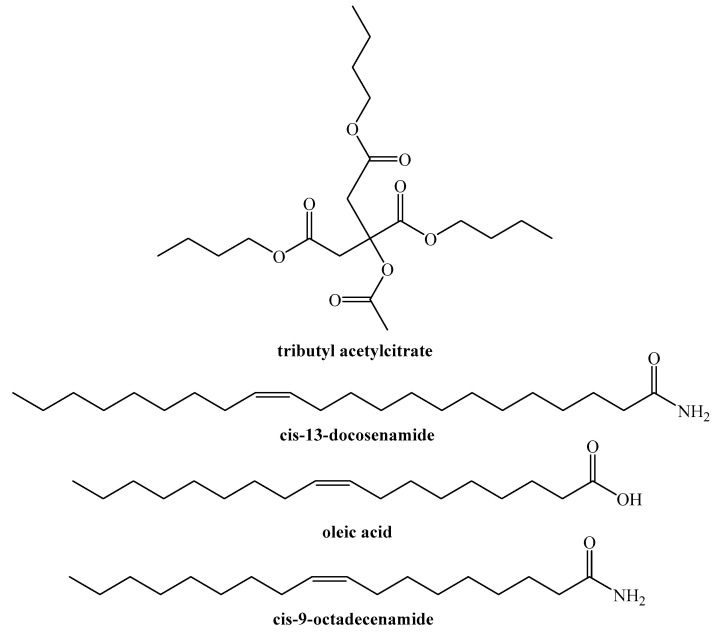
Chemical species detected by GC−MS both in *L. occidentalis* eggs and egg glue.

**Table 1 insects-14-00396-t001:** EDS elemental analyses and C/N, C/O, and N/O ratios of *L. occidentalis* eggs and egg glue components, together with those of potentially associated organic chemicals.

	C	N	O	S	Si	C/N	C/O	N/O	C*_x_*N*_y_*O*_z_*
Eggs (inner surface)	56.8	25.1	18.0	-	-	2.27	3.15	1.40	C_5_N_2_O
Eggs (outer surface)	72.0	9.7	18.3	-	-	7.42	5.57	0.53	C_10_NO_2_
Eggs glue	65.5	15.0	19.5	-	-	4.36	3.36	0.77	C_10_N_2_O_2_ or C*_x_*NO
Gelatin	50.3	17.8	25.6	-	-	2.82	1.97	0.69	C_102_[H_151_]N_31_O_39_
Lignin	46.7	9.1	27.6	1.4	-	5.13	1.69	0.33	C_18_[H_13_]N_3_O_8_S_2_
Arginine	41.4	32.2	18.4	-	-	1.28	2.25	1.75	C_6_[H_14_]N_4_O_2_
Chitin	47.2	6.9	39.4	-	-	6.84	1.20	0.18	C_6_[H_14_]NO_5_
4-Hydroxyproline	45.8	10.7	35.1	-	-	4.28	1.31	0.30	C_5_[H_9_]NO_3_
Shellac	61.4	-	30.0	-	-	-	2.04	-	C_30_[H_50_]O_11_
Hexadecanoic acid	75.0	-	12.5	-	-	-	6.00	-	C_16_[H_32_]O_2_
Oleic acid	76.6	-	14.8	-	-	-	5.17	-	C_18_[H_34_]O_2_
Pine needle	67.0	-	28.5	-	4.5	-	2.35	-	C_10_[H_11_]O_3_

## Data Availability

The data presented in this study are available on request from the corresponding author.

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
