# Peer review of "Characterization of Leptoglossus occidentalis Eggs and Egg Glue"

_insects, 2023, doi:10.3390/insects14040396_

Round 1
Reviewer 1 Report
The report may be of interest to other researchers in this field. However, there exists some flaws in the report that could be addressed to warrant further consideration after substantial re-writing.Components leading to this suggestion:
P3L141: reference other Leptoglossus rearing literature?
P3L144: extraction of what? Needle removal from the tree, or extraction of components from the needles?
P4L151: composition of the egg samples...
P4L159: egg and glue from field and from needles or from colony? If colony, what were the eggs on?
Whatever the eggs were on (organic) was that material extracted also? E.g., if on pine needles, were needles separately extracted and analyzed?
P4L171: what was injected? The sample in MeOH or another solvent? If MeOH, was the MeOH reduced in vacuo. This section needs better detail.
P8L306t is not clear if the authors are stating that extracted material is from the eggs or pine needle?
P9L326: this manuscript would be strengthened if the identified material was subjected to bioassays to determine semiochemical properties.
Author Response
Comments and Suggestions for Authors
The report may be of interest to other researchers in this field. However, there exists some flaws in the report that could be addressed to warrant further consideration after substantial re-writing.
Components leading to this suggestion:
Q1. P3L141: reference other Leptoglossus rearing literature?
Response: The conditions were identical to those specified on page 2067 of [J. Nat. Prod. 2022, 85, 2062−2070], co-authored by a member of our group. A reference has been included.
Q2. P3L144: extraction of what? Needle removal from the tree, or extraction of components from the needles?
Response: The intended meaning was, as guessed by the Reviewer, the needle removal from the trees. The text has been updated accordingly.
Q3. P4L151: composition of the egg samples...
Response: A clarification has been included, specifying ‘egg and glue’ before ‘samples’.
Q4. P4L159: egg and glue from field and from needles or from colony? If colony, what were the eggs on?
Response: As noted in subsection 2.1, “The eggs and glue samples were collected from pine needles of 2-3 years-old potted Pinus halepensis Mill. saplings (provided as ovipositional substrate) at the Forest Pathology Lab at the University of Valladolid (Palencia, Spain), where the western conifer seed bugs are reproduced and maintained in cages, in controlled laboratory conditions […]”. Nonetheless, to avoid potential confusion, we have now specified that the needles were removed “[…] from the potted pine saplings […]” in the final paragraph of that same subsection, and when the eggs and glue samples are first mentioned in subsection 2.2 (see Q3), we have further emphasized this point “[…] composition of the egg and glue samples from the abovementioned colony reared in captivity were analyzed […]”.
Q5. Whatever the eggs were on (organic) was that material extracted also? E.g., if on pine needles, were needles separately extracted and analyzed?
Response: No, P. halepensis pine needles (from the saplings provided as ovipositional substrate) were not analyzed, given that detailed phytochemical analyses for needles have been reported in the literature (see Table 2 in the recent review paper by El Omari [Journal of Ethnopharmacology, 2021, 268: 113661; DOI: 10.1016/j.jep.2020.113661] or the classic work by Anttonen et al. [Environ. Pollut. 1995, 87, 235-242, doi:10.1016/0269-7491(94)p2611-c]). Subsection 4.3 has been expanded to include a comparison with several phytochemical studies on P. halepensis (more appropriate than the reference to the work by Feng et al. on P. massoniana needles, which has now been deleted) and to discuss the presence of linoleic acid.
Q6. P4L171: what was injected? The sample in MeOH or another solvent? If MeOH, was the MeOH reduced in vacuo. This section needs better detail.
Response: Please kindly note that this information was provided in the previous paragraph: “For subsequent gas chromatography-mass spectrometry (GC-MS) analysis, 25 mg of the freeze-dried extract was dissolved in 5 mL of HPLC-grade methanol to obtain a 5 mg·mL−1 solution, which was further filtered.”, but we have now specified that it was “gradient grade, suitable as ACS-grade LC reagent, ≥99.9”, and the supplier (Sigma Aldrich).
Q7. P8L306t is not clear if the authors are stating that extracted material is from the eggs or pine needle?
Response: Please kindly note that the indicated line (“The combination of reduced penetration through the eggshell (which acts as an […]”) does not match the comment. We assume that the Reviewer is actually referring to the sentence immediately above (L301-303), “[…] Palmitic, oleic, and stearic acids have also been reported in significant amounts (18.2, 7.7, and 6.8%) in pine needle aqueous extracts [43]”. That section has been rewritten and expanded to address Q5 above, so we hope that it is now clearer to the reader.
Q8. P9L326: this manuscript would be strengthened if the identified material was subjected to bioassays to determine semiochemical properties.
Response: We agree that the suggested experiment (subjecting the identified material to bioassays to determine semiochemical properties) is interesting and would provide additional information, but we feel that it falls outside the scope of this study. We have now suggested it as a topic for further research at the end of the Discussion section of the revised manuscript. We thank the Reviewer for his/her suggestion.
Reviewer 2 Report
The manuscript “Characterization of Leptoglossus occidentalis eggs and egg glue” presents interesting and useful results that could be very helpful for the development of control against the western conifer seed bug, an important pest hurting the pine nut industry in many countries.
The paper is sloppy and not well-written, most sections are not logical and are difficult to follow. Below are my specific comments:
Line 45: remove “as regards” and start the sentence with “Some of these…”
Lines 49 – 52: you define and describe semiochemicals very well, but osmeterial secretions are only listed. I think, you should say a few words about this group of chemicals as well, most people are more familiar with the semiochemicals than with the osmeterials.
Lines 69 – 73: change to “expected of”, but overall, I am not sure what you are trying to say in this sentence.
Line 83: here and in many other places you are being too informal. Please, replace “plague” with “pest” – we know what you mean and it would be funny in a presentation, but in a scientific paper, it should be kept professional.
Line 90: replace “cycle” with “life cycle”
Line 103: environmentally friendly
Line 104: “this bug” is too informal, replace with “western conifer seed bug”
Line 109: performed poorly
Lines 111 – 112: what is leptotriene? This compound needs to be properly introduced. When writing a paper, remember, you will have some readers not fully familiar with the topic.
Line 136: add the location of the University of Valladolid (move from line 143), and replace “the bugs” with a proper name, either common or scientific
Line 147: in Petri dishes
Lines 185 – 187, 191, 208, 253 – references not found, please, fix
Overall, the Results and Discussion section is very difficult to follow. I would separate the two.
Again, this is an important study and the methodology is well described, the experiment is conducted correctly, but the style of writing makes the paper difficult to digest.
Author Response
Comments and Suggestions for Authors
The manuscript “Characterization of Leptoglossus occidentalis eggs and egg glue” presents interesting and useful results that could be very helpful for the development of control against the western conifer seed bug, an important pest hurting the pine nut industry in many countries.
The paper is sloppy and not well-written, most sections are not logical and are difficult to follow. Below are my specific comments:
Response: We thank the reviewer for his/her detailed comments, which have certainly contributed to improving the manuscript.
Q1. Line 45: remove “as regards” and start the sentence with “Some of these…”
Response: Corrected. The sentence now reads: “Some of these secretions fulfill a […]”
Q2. Lines 49 – 52: you define and describe semiochemicals very well, but osmeterial secretions are only listed. I think, you should say a few words about this group of chemicals as well, most people are more familiar with the semiochemicals than with the osmeterials.
Response: A brief explanation of what osmeterial secretions are has been included before detailing the chemicals they usually contain: “[…] (which are odoriferous liquids secreted by epidermal glands present in the prothoracic segment of papilionid larvae when the larva feels threatened) […]”.
Q3. Lines 69 – 73: change to “expected of”, but overall, I am not sure what you are trying to say in this sentence.
Response: The sentence has been slightly rephrased to improve readability: “The most complex fluid composition an insect can produce is one that comprises four key components: an aqueous portion rich in amino acids and carbohydrates, oily nanodroplets containing hydrocarbons, an emulsifier (e.g., cholesterol, monoglycerides) to stabilize the mixture, and volatile organic compounds like kairomones or pheromones.”
Q4. Line 83: here and in many other places you are being too informal. Please, replace “plague” with “pest” – we know what you mean and it would be funny in a presentation, but in a scientific paper, it should be kept professional.
Response: ‘Plague’ has been replaced with ‘pest’, as suggested.
Q5. Line 90: replace “cycle” with “life cycle”
Response: ‘Cycle’ has been replaced with ‘life cycle’.
Q6. Line 103: environmentally friendly
Response: Corrected.
Q7. Line 104: “this bug” is too informal, replace with “western conifer seed bug”
Response: ‘This bug’ has been replaced with ‘western conifer seed bug’, as indicated by the Reviewer.
Q8. Line 109: performed poorly
Response: ‘negatively’ has been replaced with ‘poorly’.
Q9. Lines 111 – 112: what is leptotriene? This compound needs to be properly introduced. When writing a paper, remember, you will have some readers not fully familiar with the topic.
Response: A clarification on what leptotriene is has been included, following the Reviewer’s recommendation: “[…] (i.e., 1S,8R,E)-9,9-dimethyl-2-methylene-5-vinylbicyclo[6.2.0]dec-4-ene, a unique sesquiterpene hydrocarbon produced by sexually-mature males of the leaffooted bugs Leptoglossus zonatus (Dallas, 1852) and L. occidentalis) […]”
Q10. Line 136: add the location of the University of Valladolid (move from line 143), and replace “the bugs” with a proper name, either common or scientific
Response: The location of the Forest Pathology Lab (viz. Palencia, Spain) has been specified, and ‘bug’ has been replaced with ‘western conifer seed bug’.
Q11. Line 147: in Petri dishes
Response: ‘on’ has been replaced with ‘in’.
Q12. Lines 185 – 187, 191, 208, 253 – references not found, please, fix
Response: Please kindly note that, in the submitted version of the manuscript, we used MS Word’s automatic numbering for figures and tables, together with cross-references in the text. Apparently, the file was edited by the Editorial Office before being sent to peer review, and the automatic numbering was removed. Consequently, cross-references were no longer valid. We have circumvented the problem by using plain text to refer to figures and tables.
Q13. Overall, the Results and Discussion section is very difficult to follow. I would separate the two.
Response: The results and discussion section has been separated into two, as advised by the Reviewer.
Q14. Again, this is an important study and the methodology is well described, the experiment is conducted correctly, but the style of writing makes the paper difficult to digest.
Response: We thank the Reviewer for his/her positive comments regarding the importance of our study and the quality of our methodology and experimental design. We also acknowledge the constructive criticism regarding the writing style of the paper. We understand that clear and concise writing is important for the effective communication of scientific research, and we have taken the feedback into account as we revised the manuscript, especially the results and discussion sections (Q13). We have strived to make these sections more accessible to the reader. We hope that the corrections made throughout the ms. and the reorganization of the original results and discussion section contents into two separate sections may have improved the overall readability of the article, making the paper more suitable for publication.
Round 2
Reviewer 1 Report
It is good to see the authors discussing the potential contamination issue with respect to the noted plasticizers.
I noticed that the authors do not mention a blank control being performed (running the entire process through with the solvents and all containers used, but no samples) to definitively eliminate contamination from the extraction and injection methods used.
Author Response
Comments and Suggestions for Authors
It is good to see the authors discussing the potential contamination issue with respect to the noted plasticizers.
Q1. I noticed that the authors do not mention a blank control being performed (running the entire process through with the solvents and all containers used, but no samples) to definitively eliminate contamination from the extraction and injection methods used.
Response: We thank the Reviewer for pointing this out. Such black controls were actually performed and allowed us to identify traces of siloxane derivatives contamination. Such ‘ghost peaks’ (which may come from a variety of sources such as inlet septa, columns, valves, solvents, and vial cap seals) were subtracted from the egg and glue sample chromatograms. These chemical species (a list is provided below) represented less than 0.9% of the peak area in the egg and glue sample chromatograms. It should be noted that none of the plasticizers (viz. butyl citrate, dibutyl itaconate, tributyl aconitate) were detected in these blank controls. Clarifications have been included in the revised version of the discussion (subsection 4.2: “[…] However, they were not present in black controls (ruling out contamination from the extraction and injection methods used) and, in additional GC-MS analyses conducted for new samples, their presence was confirmed […]”) and in subsection 2.2 (“[…] Siloxane-derivatives, which may arise from contamination from a variety of sources such as inlet septa, columns, valves, solvents, and vial cap seals, represented less than 0.9% of the peak area in the egg and glue sample chromatograms and were removed from the identified phytochemicals lists […]”).
- 1,2-Bis(trimethylsilyl)benzene
- 2,4,6-Cycloheptatrien-1-one, 3,5-bis-trimethylsilyl-
- 2-[2-[2-[2-[2-[2-(Trimethylsilyloxy)ethoxy]ethoxy]ethoxy]ethoxy]ethoxy]ethanol
- 2-[2-[2-[2-[2-[2-[2-(Trimethylsilyloxy)ethoxy]ethoxy]ethoxy]ethoxy]ethoxy]ethoxy]ethanol
- 2-[2-[2-[2-[2-[2-[2-[2-(Trimethylsilyloxy)ethoxy]ethoxy]ethoxy]ethoxy]ethoxy]ethoxy]ethoxy]ethanol
- 2-[2-[2-[2-[2-[2-[2-[2-[2-(Trimethylsilyloxy)ethoxy]ethoxy]ethoxy]ethoxy]ethoxy]ethoxy]ethoxy]ethoxy]ethanol
- 2-[2-[2-[2-[2-[2-[2-[2-[2-[2-(Trimethylsilyloxy)ethoxy]ethoxy]ethoxy]ethoxy]ethoxy]ethoxy]ethoxy]ethoxy]ethoxy]ethanol
- 3,4-Dimethylbenzoic acid, tert-butyldimethylsilyl ester
- 4-Methyl-2-trimethylsilyloxy-acetophenone
- 5-Methyl-2-trimethylsilyloxy-acetophenone
- Arsenous acid, tris(trimethylsilyl) ester
- Cyclohexasiloxane, dodecamethyl-
- Cyclopentasiloxane, decamethyl-
- Cyclotetrasiloxane, octamethyl-
- Propanoic acid, 2-methyl-, tert-butyldimethylsilyl ester
- Silanamine, N-[2,6-dimethyl-4-[(trimethylsilyl)oxy]phenyl]-1,1,1-trimethyl-
- Silane, dimethyl-
- Silane, trimethyl(phenylmethoxy)-
- Silanediol, dimethyl-
- Tetrasiloxane, decamethyl-
- Trimethylsilyl 3-methyl-4-[(trimethylsilyl)oxy]benzoate
- Tris(tert-butyldimethylsilyloxy)arsane